# The Impact of Aqueous Extracts of *Verbesina sphaerocephala* and *Verbesina fastigiata* on Germination and Growth in *Solanum lycopersicum* and *Cucumis sativus* Seedlings

**Ana Paulina Velasco-Ramírez** [1,*] , **Alejandro Velasco-Ramírez** [1], **Rosalba Mireya Hernández-Herrera** [2], **Jesus Ceja-Esquivez** [1], **Sandra Fabiola Velasco-Ramírez** [3], **Ana Cristina Ramírez-Anguiano** [3] **and Martha Isabel Torres-Morán** [1]

1 Departamento de Producción Agrícola, Centro Universitario de Ciencias Biológicas y Agropecuarias (CUCBA), Universidad de Guadalajara, Zapopan 45010, Mexico; alejandro.velasco@academicos.udg.mx (A.V.-R.); jesus.cesquivez@alumnos.udg.mx (J.C.-E.); isabel.torres@academicos.udg.mx (M.I.T.-M.)

2 Departamento de Botánica y Zoología, Centro Universitario de Ciencias Biológicas y Agropecuarias (CUCBA), Universidad de Guadalajara, Zapopan 45010, Mexico; rosalba.hernandez@academicos.udg.mx

3 Departamento de Química, Centro Universitario de Ciencias Exactas e Ingenierías (CUCEI), Universidad de Guadalajara, Guadalajara 44430, Mexico; sandra.vramirez@academicos.udg.mx (S.F.V.-R.); ana.ranguiano@academicos.udg.mx (A.C.R.-A.)

\* Correspondence: paulina.velasco@academicos.udg.mx; Tel.: +52-1-333-777-1150 (ext. 33141)

**Abstract:** The use of extracts derived from different plants has gained popularity as an alternative option to manage weeds and support phenological development in plants. One of the main problems facing agricultural production is the intensive application of chemical fertilizers that harm the environment. This study investigated the biostimulant effects of the aqueous extracts of *Verbesina sphaerocephala* and *Verbesina fastigiata* on the germination and growth of tomato (*Solanum lycopersicum*) and American cucumber (*Cucumis sativus*) seedlings. The aqueous extracts of both species of *Verbesina* were tested at concentrations of 0.5 and 1%. Seed germination and seedling development under nursery conditions were evaluated. The seed germination percentage was reduced in all treatments with the extracts of both *Verbesina* species when compared to that of the control treatment, which was considered an allelopathic effect. However, seedling growth in pots showed different behavior, with the extracts beneficially affecting certain agronomic variables, such as root and leaf length. The results suggest potential uses for *V. sphaerocephala* and *V. fastigiata* extracts, although a greater range of action may be experienced through the effects of certain molecules on weed growth. Additional studies with chemical approaches are necessary to better direct the uses and applications of *Verbesina* biomolecules, as they also functioned as growth promoters.

**Keywords:** botanical extracts; allelopathy; sustainability; capitaneja; horticulture; growth biostimulant

## 1. Introduction

The main challenge facing modern agricultural science is finding ways to maintain food production in order to meet the needs of the growing world population without compromising resources for future generations [1]. Currently, agriculture production has reached a critical level given its multiple negative effects, such as irreversible global climate change, the loss of many ecosystem services, and the introduction of harmful contaminants and pollutants into terrestrial and marine habitats [2]. The main agricultural pollutants that threaten the environment are pesticides, nitrates from nitrogen-rich fertilizers, and phosphorus [3,4]. The problems surrounding environmental pollution and herbicide resistance have forced scientists to develop alternative weed management, germination, and plant growth methods [5–7]. Over the last two decades, plant-derived extracts have gained popularity as alternatives to traditional fertilizers to manage weeds and support

phenological development in plants [8]. This is in part because plant extracts result in lower levels of pollution and are more biodegradable than traditional herbicides and synthetic agrochemicals [9]. Through the phenomenon of allelopathy, these extracts have shown notable potential to both retard or promote seed germination and seedling growth [10,11].

Allelopathy is a biological phenomenon in which a plant influences the growth and development of other plants by producing phytochemicals and releasing them into the surrounding environment [12,13]. Allelochemicals are manufactured as by-products or secondary metabolites in the stems, leaves, roots, flowers, inflorescences, fruits, and seeds of plants, although allelochemical production in leaves has been the most commonly documented [14]. Allelochemicals can have beneficial or detrimental effects on target plants. Within the context of agricultural practices, allelopathy can be considered to be a process of chemical interference between crops and weeds that ultimately affects plant production.

Several allelochemicals have been aqueously extracted from various botanical species and have not been classified as either germination inhibitors or retardants, but as growth promoters. For example, the aqueous extracts of three wild herbs (*Achatherum splendens*, *Artemisia frigida*, and *Stellera chamaejasme*) were found to allelopathically inhibit lettuce germination, but promote seedling growth when applied in low concentrations [15]. In addition, aqueous extracts of fig leaves (*Ficus carica* L.) were found to affect germination, but promote growth in three medicinal species (Mint, Dandelion, and Woad) [16].

Extracts of *Verbesina sphaerocephala* and *Verbesina fastigiata* can be combined to create a botanical strategy to reduce the current dependence on agrochemicals [17]. The agricultural potential of *V. sphaerocephala* as a bioremediator, plant growth promoter, and even a pharmaceutical has already been reported by Hernández-Pérez [18]. This author demonstrated that the development of American cucumber plants (*Cucumis sativus*) was supported by the application of aqueous extracts from *V. sphaerocephala* leaves, which reduced damage caused by the nematode *Meloidogyne incognita*. In addition, *V. fastigiata* has been found to promote growth in *Fragaria* × *ananassa* [19]. Both *V. sphaerocephala* and *V. fastigiata* belong to the Asteraceae family and are endemic to western Mexico (i.e., the states of Jalisco, Michoacán, Nayarit, Guanajuato, and Guerrero) [20], where they are known by their common names of *capitana* and *arnica capitaneja*, respectively. However, the use of *V. sphaerocephala* or *V. fastigiata* is not widespread and only taxonomic, ethnobotanical information on either species is available [21], with the exception of recent chemical studies of *V. sphaerocephala* [22,23]. Therefore, we propose that the aqueous extracts of both species function as growth promoters. The aqueous extracts of both species can be made by the same producer and thus used as alternatives to minimize the use of herbicides and agrochemicals in crops of commercial interest, such as tomato (*Solanum lycopersicum*) and American cucumber (*C. sativus*).

Tomatoes are the most widely produced vegetable in the world, with annual production totaling 186,821,216 t. In Mexico, the ninth largest producer of tomatoes in the world, 470,527 t of tomatoes were produced in 2020. In the same year, the main tomato-producing states in Mexico were Sinaloa (764,435 t), San Luis Potosí (391,719 t), Michoacán (280,478 t), Zacatecas (189,319 t), and Jalisco (175,999 t) [24]. Cucumber is the fourth most cultivated vegetable worldwide, and 1.7 million t were produced in 2016 [25]. Mexico is the fifth largest cucumber producer in the world, with its production totaling 826,485 tons. The Mexican states that produce the most cucumbers are Sinaloa (268,878 t), Sonora (152,457 t), and Michoacán (67,653 t). Jalisco, which occupies tenth place in cucumber production in Mexico, generated 20,454 t in 2020 [24].

The main problem facing tomato and cucumber production in Mexico is the intensive application of chemical fertilizers, which reduce soil quality and threaten agricultural systems [26]. Therefore, the objective of this study was to evaluate the effects of the aqueous extracts of *V. sphaerocephala* and *V. fastigiata* on the germination of tomato (*S. lycopersicum*) and American cucumber (*C. sativus*) seedlings and to determine if any biostimulant effects could be observed.

## 2. Materials and Methods

### 2.1. Study Site

This study was developed in the University Center for Biological and Agricultural Sciences (CUCBA). The first stage of the project, which consisted of the germination of seeds in Petri dishes, was carried out in the Molecular Markers Laboratory. The second stage, which consisted of seedling adaptation, was conducted in the school nursery. Both the laboratory and school nursery belong to the University of Guadalajara, which is located in Zapopan, Jalisco (20°45′ N and 103°31′ W) at ~1510 m. a.s.l., although the school nursery is found at ~1650 m. a.s.l.

### 2.2. Preparation of the Aqueous Extracts of V. sphaerocephala and V. fastigiata

Healthy (asymptomatic) young leaves (vegetative phase) of wild *V. sphaerocephala* were collected from the hills surrounding the university in the community of San Martin de las Flores in the municipality of San Pedro Tlaquepaque, Jalisco (20.585278° N, 103.282778° W; 1540 m. a.s.l.). Wild *V. fastigiata* was collected in the spring forest of the municipality of Zapopan, Jalisco (20.28° N, 103.50° W). The leaves were dried at room temperature in the laboratory (~26 °C) and then pulverized in an 80,335 blade mill (Hamilton Beach®, Glen Allen, USA). Subsequently, 100 g leaves (dry weight) was poured into 1 L distilled water and constantly stirred for 15 min. The leaves were then autoclaved at 121 °C for 1 h at 1.2 kg cm$^2$. The hot extracts were passed through Whatman No. 40 filter paper (Maidstone, UK) and stored in glass jars at 4 °C. The liquid extracts of *V. sphaerocephala* and *V. fastigiata* were designated as mother solutions, and their pH and electrical conductivity (EC, dS m$^{-1}$) were measured. Finally, the color of the extracts was visually determined. All parameters were evaluated in triplicate.

### 2.3. Chemical Characterization of the Aqueous Extracts of V. sphaerocephala and V. fastigiata

An HI83325 multiparametric photometer (HANNA® Instruments, Woonsocket, RI, USA) was used to characterize the aqueous extracts of both *Verbesina* species. The pH, electrical conductivity (EC), and concentrations of ammonium nitrate ($NH_4NO_3$), Ca, Mg, nitrate, phosphate ($PO_4^{3-}$), and K were determined following the instructions of the manufacturer. The chemical parameters were determined in triplicate in each aqueous extract.

### 2.4. Selection of Crop Plants

Certified SEMINIS® brand tomato (*S. lycopersicum*, hybrid tomate v. DRT 8551) seeds and commercial cucumber seeds (*C. sativa* v. Poinsett 76) were selected with uniform size, color, and weight.

### 2.5. Bioassays under Laboratory Conditions

#### 2.5.1. Moisture Content of Seeds

Three 100-seed samples of each species (*S. lycopersicum* and *C. sativa*) were collected and used to determine moisture content (MC). The dry weight of the seeds was obtained with an analytical balance after oven drying at 130 °C for 1 h [27]. The MC was determined using Equation (1):

$$MC = \frac{Fresh\ weight - Dry\ weight}{Dry\ weight} \times 100 \tag{1}$$

#### 2.5.2. Imbibition

Seeds of *S. lycopersicum* and *C. sativa* were placed in Petri dishes with a cotton support and filter paper and moistened with distilled water. Then, the Petri dishes were kept in a chamber with a controlled environment (constant temperature of 28 ± 1 °C) in the dark. Three repetitions with 20 seeds each (*n* = 120) were included for each species. Weight was recorded every 4 h for 48 h and a final weight was taken at 72 h. The imbibition percentage was determined as the increase in seed weight due to water absorption with respect to the initial weight.

### 2.6. Seed Germination

Prior to germination, the cucumber seeds were disinfected using 30% chlorine (150 mL for every 500 mL of distilled water). The seeds were soaked in the disinfectant solution for 15 min under agitation and then washed three times with distilled water.

Germination was observed each day for 8 days, according to the methods of the Association of Official Seed Analysts [28]. Germination was evaluated in five 20-seed groups per species per treatment [28]. Germination bioassays were performed according to the methods of Hernández-Herrera et al. [26]. Tomato and cucumber seeds were placed on Whatman No. 5 filter paper in sterilized 90-mm Petri dishes, and then treated with either $5 \text{ mL}^{-1}$ distilled water (control) or one of two concentrations (i.e., 0.5 and 1.0%) of the aqueous extracts of *V. sphaerocephala* or *V. fastigiata*. The plates were incubated at $25 \pm 1\ ^{\circ}\text{C}$ under cool white fluorescent light ($50\ \mu\text{mol m}^{-2}\,\text{s}^{-1}$ photosynthetic photon flux density) with a photoperiod of 16 h light/8 h dark. The final germination percentage (GP) was calculated on the eighth day after sowing, according to the methods of the Association of Official Seed Analysts [28], and was expressed with Equation (2):

$$GP = \frac{Number\ of\ germinated\ seeds\ at\ the\ final\ count}{Total\ number\ of\ seeds\ sets\ in\ the\ bioassay} \times 100 \tag{2}$$

To evaluate the mean germination time (MGT), 200 seeds were sown between two germination filter papers (four replicates with 50 seeds/replica) per vegetable species and incubated in a seed germinator in a dark room at a constant temperature of 25 °C. Germination counts were made every 24 h and the mean germination time (MGT) [29] was calculated with Equation (3):

$$MGT = \frac{\Sigma nd}{\Sigma n} \tag{3}$$

where *n* is the number of newly germinated seeds per day and *d* is the number of days from the start of the germination test. The vigor index (SVI) was determined according to the methods of Rathinapriya et al. [29] with Equation (4):

$$SVI = Seedling\ vigor \times GP, \tag{4}$$

where seedling vigor is the root and shoot length in cm, and *GP* is the germination percentage.

### 2.7. Ex Vitro Adaptation of Tomato and Cucumber Seedlings

After 12 days of seed germination and primary root development, tomato and cucumber seedlings were acclimated to ex vitro conditions in the school nursery. Twenty of the best seedlings per treatment of each species were selected (i.e., 100 plants per species and 20 per treatment). The seedlings were acclimated in 200, 3-inch plastic pots filled with a previously sifted mixture of jal substrate (organic substrate of volcanic origin), coconut powder, and oak leaves in equal parts (33.3%). All pots were irrigated at field capacity. After nine days of acclimation, the first fertilization directed to the ground was conducted by applying 0.5 mL/L of Poliquel Zinc®, Arysta Pilatus®, and MAP 0.5 g/L. Fertilization was performed three times per week. On the same day as the start of fertilization, the first application of the aqueous extracts of *V. sphaerocephala* and *V. fastigiata* was conducted, and 5 mL/L was applied directly to the soil once a week.

### 2.8. Growth Parameters in Tomato and Cucumber Seedlings

The growth parameters of leaf length, root length, leaf area, and root area were measured after 30 days of growth. The parameters were calculated according to the methods of Hernández-Herrera et al. [26].

### 2.9. Statistical Analysis

In all cases, normality and homoscedasticity tests were first used to evaluate the data. To compare the means of multiple groups or treatments, a one-way analysis of

variance (ANOVA) and the least significant difference (LSD) multiple comparison test ($p = 0.05$) were used. All statistical analyses were performed with Statgraphics® Centurion XV for Windows.

## 3. Results

The germination percentage was reduced in all treatments that included the application of the aqueous extracts of both *Verbesina* species when compared to that of the control treatment. However, the beneficial effects of the application of the liquid extracts were observed in the growth of potted seedlings for certain agronomic variables. As such, the germination percentage was used to evaluate the allelopathic potential of the aqueous extracts of both *Verbesina* species.

### 3.1. Chemical Characterization of the Aqueous Extracts of V. sphaerocephala and V. fastigiata

The extract colors of both *Verbesina* species were an intense green, and the pH values of the extracts were slightly alkaline. The chemical parameters of each extract are shown in Table 1. The extracts of both species contained differing amounts of elements and compounds, although neither nitrate nor K compounds were detected in either extract. The ammonia values were higher in the extract of *V. fastigiata* than that of *V. sphaerocephala*, whereas Ca, Mg, and phosphate were higher in the *V. sphaerocephala* extract (Table 1).

**Table 1.** Chemical properties of the aqueous extracts of *Verbesina sphaerocephala* and *Verbesina fastigiata*.

| Parameter | *V. sphaerocephala* | *V. fastigiata* |
|---|---|---|
| pH | 8.33 | 8.24 |
| EC (dSm$^{-1}$) | 1 a 1.3 | 1 a 1.3 |
| Ammonia (NH$_4$NO$_3$; mg/L) | 0.3 | 0.9 |
| Ca (mg/L) | 48 | 10 |
| Mg (mg/L) | 87 | 5 |
| Nitrate (mg/L) | Nd | Nd |
| Phosphate (mg/L) | 11.1 | 6.4 |
| K (mg/L) | Nd | Nd |

EC = Electric conductivity; Nd = not detected.

### 3.2. Effects of the Aqueous Extracts of V. sphaerocephala and V. fastigiata on the Germination of Tomato and Cucumber Seeds

The germination of the tomato and cucumber seeds was similar among treatments in which the aqueous extracts of the *Verbesina* species were applied. When the aqueous extract of *V. sphaerocephala* was applied to tomato seeds, no significant effects on the seed germination percentage at any concentration (0.5 and 1%) with respect to that of the control treatment were evident ($15.2 \pm 2.6$, $13.4 \pm 0.4$, respectively; Table 2), and a delayed effect was evident. These results were associated with what was observed with the MGT. On the fourth day of treatment with the 0.5% aqueous solution, radicle development began and resulted in 30% germination, with 92% germination being observed on the eighth day. However, the seeds treated with the 1% aqueous solution often did not develop, as the MGT on the eighth day was 23% (Table 2; Figure 1a,b).

It is worth mentioning that the germination percentage of the treatments in which the aqueous extract of *V. fastigiata* was applied to tomato seeds also did not show significant differences in the germination percentage at any concentration (0.5 and 1%; $17.2 \pm 3.2$, $20.0 \pm 1.4$, respectively) with respect to that of the control treatment (Table 2). However, it is evident that germination was delayed in the seeds of these treatments, as shown by the MGT (21% on the fourth day with the 0.5% aqueous solution and 95% germination on the eighth day). When treated with the 1% aqueous solution, a germination percentage of 77% was obtained on the eighth day, with delayed germination also evident (Table 2; Figure 1a,c).

**Table 2.** Effect of aqueous extracts of *Verbesina sphaerocephala and Verbesina fastigiata* on germination percentage, mean germination time, and seed vigor index.

| (%) | Germination (%) | | Mean Germination Time (MGT; days) | | Vigor Index (SVI) | |
|---|---|---|---|---|---|---|
| | Tomato | Cucumber | Tomato | Cucumber | Tomato | Cucumber |
| Control | 58.3 ± 3.3 [c] | 59.6 ± 4.0 [d] | 6.3 ± 0.08 [d] | 7.6 ± 0.09 [d] | 67 ± 40.07 [c] | 76 ± 1.1 [c] |
| V.s 0.5 | 15.2 ± 2.6 [a] | 20.0 ± 2.0 [b] | 4.1 ± 0.30 [b] | 5.4 ± 0.06 [c] | 10 ± 0.4 [a] | 36 ± 2.0 [a] |
| V.s 1.0 | 13.4 ± 0.4 [a] | 23.5 ± 0.5 [b] | 0.9 ± 0.1 [a] | 1.0 ± 0.04 [a] | 0.6 ± 0.3 [a] | 34 ± 1.8 [a] |
| V.f 0.5 | 17.2 ± 3.2 [a] | 27.2 ± 3.2 [a] | 4.0 ± 0.20 [b] | 4.4 ± 0.25 [b] | 14 ± 0.5 [b] | 40 ± 0.7 [b] |
| V.f 1.0 | 20.0 ± 1.4 [b] | 27.8 ± 1.2 [a] | 5.6 ± 0.50 [c] | 5.3 ± 0.50 [c] | 12 ± 0.5 [b] | 43 ± 1.6 [b] |

Values are mean ± standard error ($n = 200$). Means followed by the same letter within a column are not significantly different according to the least significant difference (LSD) multiple range test ($p \le 0.05$).

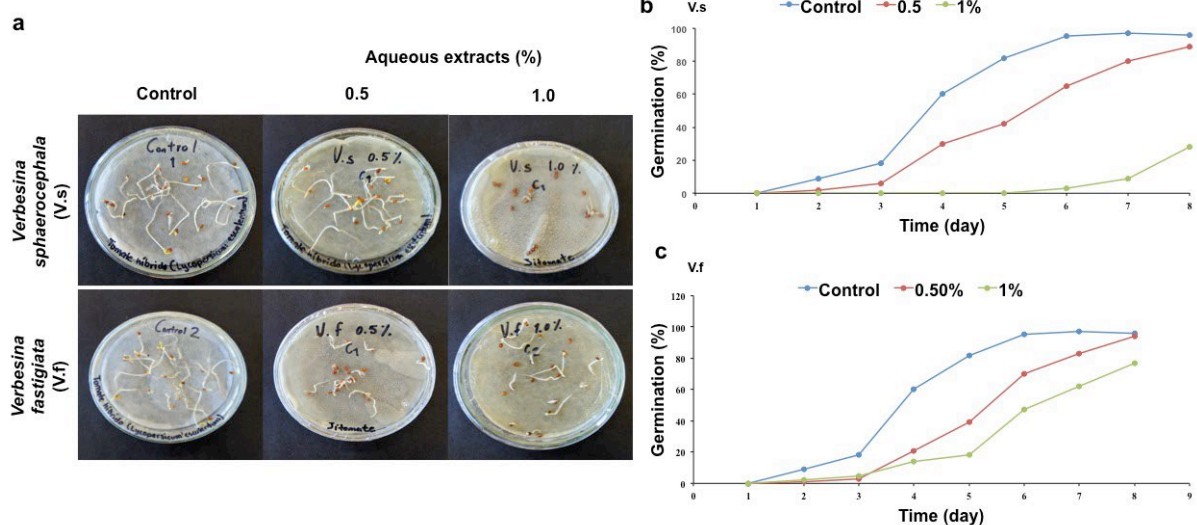

**Figure 1.** Germination of tomato (*Solanum lycopersicum*) seeds. (**a**) Application of the aqueous extracts of *Verbesina sphaerocephala* and *Verbesina fastigiata* at concentrations of 0.5 and 1.0%. (**b**) Germination percentages (GP) obtained with aqueous extracts of *V. sphaerocephala* ($n = 100$, $p = 0.05$) and (**c**) *V. fastigiata* ($n = 100$, $p = 0.05$).

The effects of the *V. sphaerocephala* extracts on the germination of cucumber seeds were similar with respect to the effects observed on the germination of tomato seeds. The extract concentrations of 0.5 and 1% resulted in germination percentages (20.0 ± 2.0 and 23.5 ± 0.5, respectively) that were not significantly different with respect to that of the control (Table 1). On the fourth and eighth days with the extract concentration of 0.5%, the MGT values were 30% and 82%, respectively, while the 1% aqueous solution resulted in a delay in germination, yielding an MGT value of 23% on day 8 (Table 2, Figure 2 a,b).

The results obtained with the application of the aqueous extracts of *V. fastigiata* in cucumber seeds were similar to those obtained with tomato seeds. The extract concentrations of 0.5 and 1% did not result in significant differences in the germination percentage (27.2 ± 3.2 and 27.8 ± 1.2, respectively) compared to that of the control; however, the MGT reflected similar germination percentages of 20% and 95% on the fourth and eighth days, respectively, with the extract concentration of 0.5%. A germination percentage of 75% was obtained on the eighth day with the extract concentration of 1%, although a delay in germination was also evident (Table 2, Figure 2a,c).

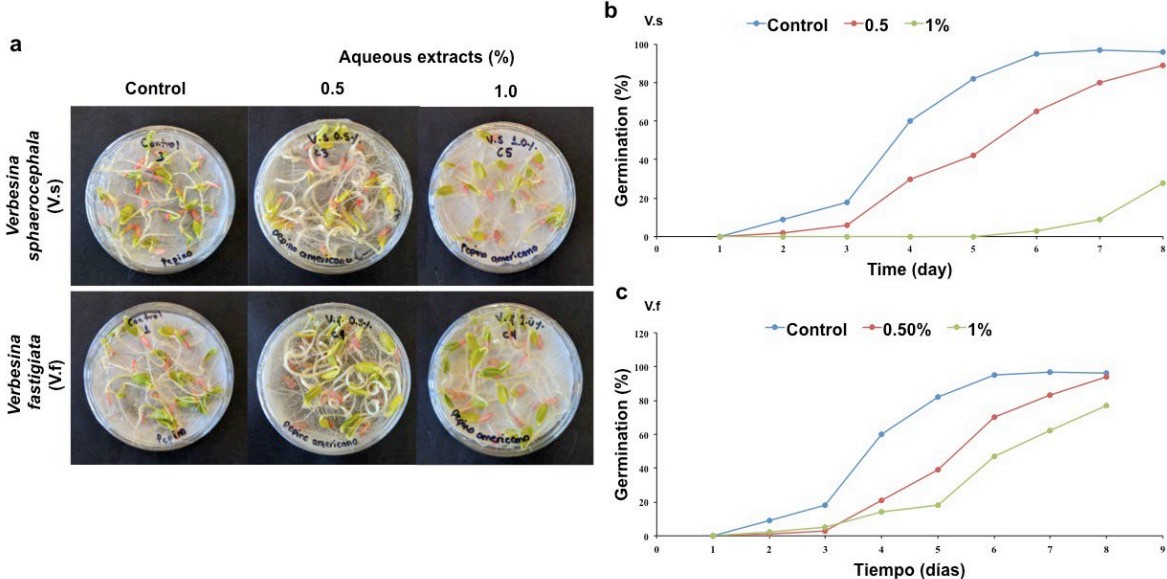

**Figure 2.** Germination of cucumber (*Cucumis sativus*) seeds. (**a**) Application of the aqueous extracts of *Verbesina sphaerocephala* and *Verbesina fastigiata* at concentrations of 0.5 and 1.0%. (**b**) Germination percentages (GP) obtained with the aqueous extracts of *V. sphaerocephala* ($n = 100$, $p = 0.05$) and (c) *V. fastigiata* ($n = 100$, $p = 0.05$).

In all treatments with the application of the aqueous extracts of both *Verbesina* species, the SVI values were significantly different from those of the control treatment (67 ± 40.07 and 76 ± 1.1, respectively; Table 2).

### 3.3. Effects of the Aqueous Extracts of V. sphaerocephala and V. fastigiata on the Development of Tomato and Cucumber Plants

#### 3.3.1. Sheet Length

The *V. sphaerocephala* extract applied at a concentration of 0.5% to the substrate resulted in an average tomato plant length of 108 mm, while the tomato plants treated with the extract at a concentration of 1% showed greater seedling lengths. The leaves of the tomato plants treated with the 1% *V. sphaerocephala* extract measured an average of 117 mm, whereas those of the control plants measured 110 mm on average. The data showed a polynomial trend, reflecting fluctuation with regard to gains in leaf length, especially in treatments with the application of the extract at a concentration of 1% (Figure 3a). With the application of the *V. fastigiata* extract, average leaf lengths of 90 and 98 mm were obtained with the extract concentrations of 0.5% and 1%, respectively. The data also showed a polynomial trend in which it was evident that the control plants showed greater development than those in the other treatments. The plants treated with the extract concentration of 0.5% showed lower growth with respect to those treated with the 1% aqueous solution (Figure 3a).

In cucumber seedlings, average leaf lengths of 60 mm and 80 mm were obtained in plants treated with the extracts of *V. sphaerocephala* at concentrations of 0.5% and 1%, respectively. The trend in the data was polynomial, and a greater average leaf length of 99 mm was observed in the control plants with respect to those treated with the extract. In these plants, leaf length was lower in plants treated with the 0.5% aqueous solution compared to those treated with the 1% aqueous solution (Figure 3b). However, with the application of the *V. fastigiata* extract, the greatest leaf lengths were observed in plants treated with the 0.5% aqueous solution (average of 123 mm) compared to those of plants treated with the 1% aqueous solution (60 mm). This reflects a trend in the data of a second-degree polynomial, with a maximum in leaf length obtained with the application of the 0.5% aqueous solution and a minimum obtained with the application of the 1% aqueous solution (Figure 3b).

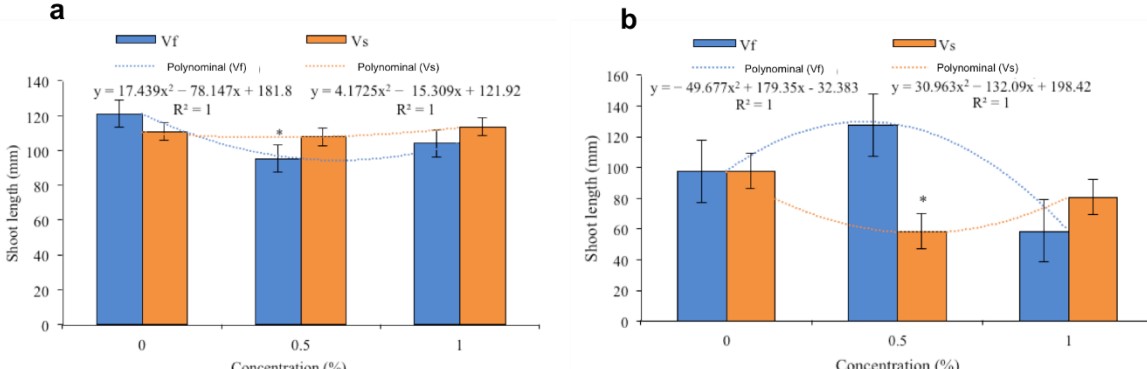

**Figure 3.** Effects of the aqueous extracts of *Verbesina sphaerocephala* and *Verbesina fastigiata* at different concentrations on leaf length in (**a**) *Solanum lycopersicum* and (**b**) *Cucumis sativus*. Columns denoted by an asterisk (*) are statistically different from the control. The values represent the average (*n* = 20 seedlings); the bars represent the standard error.

### 3.3.2. Root Length

The *V. sphaerocephala* extract applied at a concentration of 0.5% to the substrate resulted in an average root length of 90 mm in tomato seedlings, while greater root lengths were observed in tomato plants treated with the extract at 1% (119 mm). Both of these values were greater than the average root length of the control plants of 82 mm. A linear trend was observed in the data indicating that the root length increased according to the concentrations of the extract (Figure 4a). The application of the *V. fastigiata* extract at a concentration of 0.5% resulted in an average root length of 83 mm in tomato plants, while the extract concentration of 1% resulted in an average root length of 99 mm. The control plants had a lower average root length (80 mm). In this case, the data also showed a linear trend, with the concentration at 1% being significantly different to that of the control (Figure 4a).

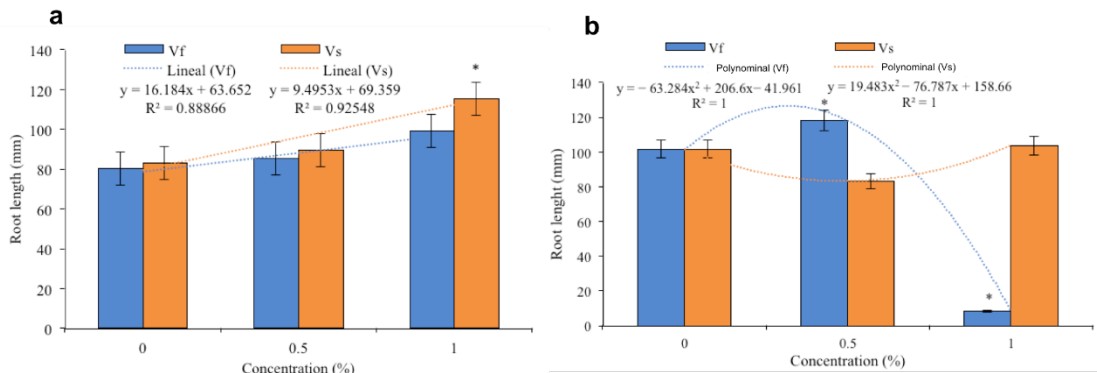

**Figure 4.** Effects of the aqueous extracts of *Verbesina sphaerocephala* and *Verbesina fastigiata* at different concentrations on root length in (**a**) *Solanum lycopersicum* (**b**) and *Cucumis sativus*. Columns denoted by an asterisk (*) are statistically different from the control. The values represent the average (*n* = 20 seedlings); the bars represent standard error.

The *V. sphaerocephala* extracts applied to the substrate in concentrations of 0.5 and 1.0% resulted in average root length values in cucumber seedlings of 80 and 103 mm, respectively, although these values were similar to that of the control plants (100 mm; Figure 4b). The application of the *V. fastigiata* extract at a concentration of 0.5% resulted in growth (120 m) that was significantly different from that of the control. With the application of the extract at a concentration of 1%, root growth was only 8 mm, which was the lowest value observed. Root growth with the application of the *V. fastigiata* extract at a concentration of 0.5% was similar to that observed with the 1% concentration of the *V. sphaerocephala* extract. The data

showed a polynomial trend, as a notable decrease in root growth was observed with the 1% aqueous solution (Figure 4b).

### 3.3.3. Leaf Area

The *V. sphaerocephala* extract applied at concentrations of 0.5 and 1.0% to the substrate resulted in unfavorable outcomes with respect to leaf area in tomato seedlings, as the seedlings had leaf areas of less than 10,000 mm$^2$. In contrast, the average leaf area of the control treatment was much higher (25,000 mm$^2$). The data showed a linear trend, with a greater leaf area in the control treatment that declined with extract concentrations of 0.5% and 1% (Figure 5a). The opposite was observed with the application of the *V. fastigiata* extract at 0.5%, with an average leaf area of 40,000 mm$^2$ being obtained. However, with the application of the extract at a concentration of 1%, the average leaf area (60,000 mm$^2$) was significantly greater than that of the control (Figure 5a).

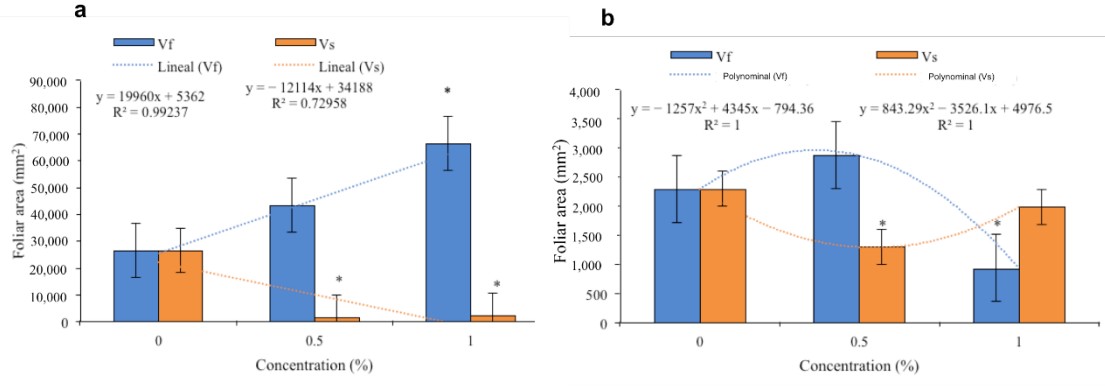

**Figure 5.** Effects of the aqueous extracts of *Verbesina sphaerocephala* and *Verbesina fastigiata* at different concentrations on leaf area in (**a**) *Solanum lycopersicum* and (**b**) *Cucumis sativus*. Columns denoted by an asterisk (*) are statistically different from the control. The values represent the average (*n* = 20 seedlings); the bars represent the standard error.

The application of *V. sphaerocephala* extracts at 0.5 and 1.0% resulted in leaf areas of 1300 mm$^2$ and 2000 mm$^2$ in cucumber seedlings; however, both concentrations resulted in leaf areas that were lower than that of the control treatment (2300 mm$^2$). The data showed a polynomial trend, with elevated values in the control treatment that decreased with the application of the extract at a concentration of 0.5% and increased with the application of the 1% aqueous solution (Figure 5b). When the *V. fastigiata* extract was applied at a concentration of 0.5%, an average leaf area of 28,000 mm$^2$ was obtained, which was significantly greater than the values obtained with the application of the extract at a concentration of 1% (700 mm$^2$) and the control (2300 mm$^2$), which were not significantly different from one another. The trend in the data was that of a second-degree polynomial, with a maximum in leaf area occurring with the 0.5% aqueous solution and a minimum value occurring with the 1% aqueous solution (Figure 5b).

### 3.3.4. Root Area

The *V. sphaerocephala* extract applied at a concentration of 0.5% to the substrate resulted in a root area of 10,000 mm$^2$ in tomato plants, which was significantly greater than those of the 1% aqueous solution (1000 mm$^2$) and control treatment (1000 mm$^2$). The trend in the data was that of a second-degree polynomial, with a maximum value obtained with the 0.5% aqueous solution and a minimum value obtained with the 1% aqueous solution (Figure 6a). The application of the *V. fastigiata* extract did not result in significant differences among root areas at any concentration. With both the 0.5% and 1% aqueous solutions, root areas of less than 1000 mm$^2$ were obtained. The trend in the data was linear and the results were not significant (Figure 6a).

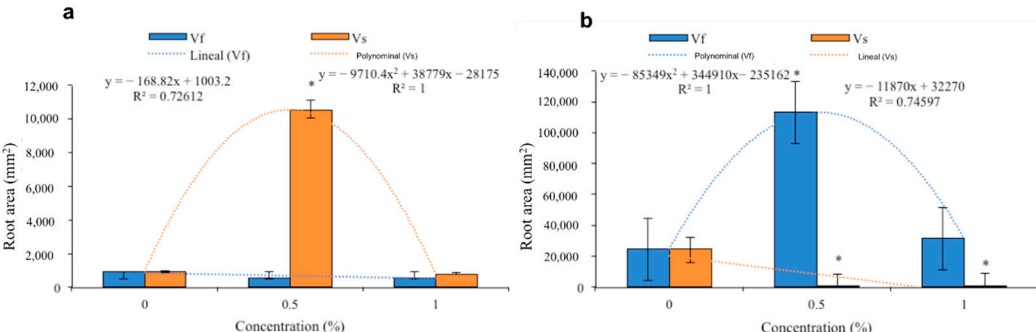

**Figure 6.** Effects of the aqueous extracts of *Verbesina sphaerocephala* and *Verbesina fastigiata* at different concentrations on root area in (**a**) *Solanum lycopersicum* and (**b**) *Cucumis sativus*. Columns denoted by an asterisk (\*) are statistically different from the control. The values represent the average (*n* = 20 seedlings); the bars represent the standard error.

The application of the *V. sphaerocephala* extract to the substrate at concentrations of 0.5% and 1% did not result in significantly different values of root area between the two treatments. However, the average root area of the control treatment was 20,000 mm$^2$ (Figure 6b). The trend in the data was linear with a slight slope (Figure 6b). The application of the *V. fastigiata* extract at a concentration of 0.5% resulted in a root area of 100,000 mm$^3$, which was significantly greater than that obtained with the application of the extract at a concentration of 1% (30,000 mm$^2$). The lowest values were obtained in the control treatment (20,000 mm$^2$). The trend in the data was that of a polynomial, with a maximum value obtained with the 0.5% concentration (Figure 6b).

## 4. Discussion

The chemical properties of the aqueous extracts of both Verbesina species were compared with those of a previous study that focused on strawberry (*Fragaria x ananassa*) cultivation [20]. We did not detect nitrates and K compounds were not detected in the extracts, which could be due to the fact that the extracts are considered organic, and that these nutrients could be absorbed and assimilated by plants through enzymatic activation derived from various proteins contained within the extract. The species of *Verbesina sphaerocephala* has a high content of organic matter, carbohydrates, and proteins [19]. However, results may vary according to the phenological state of the leaves used to obtain the extracts and may even vary based on the analytical method. It is important that the collection is from the young leaves of the plant so that the chemical analyses are more specific in detecting chemical elements as well as secondary metabolites.

### 4.1. Effect of V. sphaerocephala and V. fastigiata Extracts on the Germination Rate of S. lycopesicum and C. sativus

The results of this study seem to indicate that the aqueous extracts of both *Verbesina* species reduced and inhibited (*p* < 0.05) the germination percentages of *S. lycopersicum* and *C. sativus* seeds. Therefore, it is very likely that both *Verbesina* species contain secondary metabolites with allelopathic functions. Various weeds and wild plants that grow and develop in cultivated fields can contain allelopathic substances that affect seed germination [30]. Various studies, i.e., [15,16,31], have indicated that most allelopathic substances, such as secondary metabolites, are synthesized through the metabolic shikimic acid and isoprene pathways. The known allelopathic substances mainly include phenols, quinones, karinas, flavonoids, terpenes, sugars, glycosides, alkaloids, and non-protein amino acids [32]. Given that *V. sphaerocephala* has been found to contain large amounts of phenolic compounds and alkaloids [17,19], it is likely that *V. fastigiata* also contains these compounds, as it is also of the same genus.

The allelopathic response that caused the extracts of *V. sphaerocephala* and *V. fastigiata* to inhibit seed germination in the plants in this study points mainly to the quantity and

quality of phenolic compounds, as indicated in a phytochemical study carried out on the aqueous extracts of the species *V. sphaerocephala*, in which the authors emphasize its high content of phenolic compounds [19]. While low concentrations can favor development, high concentrations inhibit the same. The concentrations evaluated in this study inhibited and/or delayed root development. It has been found that the accumulation of phenolic compounds in the seed coat can influence the germination rate and certain alkaloids can behave as allelopathic compounds [33].

*Verbesina* species belong to the Asteraceae family and, as such, have been reported to be potential sources of lactones (mainly sesquiterpenes), which function as secondary metabolites and in turn show a wide variety of biological activities, while being useful precursors in the synthesis of natural products and compounds with insecticidal and parasiticidal activity [34]. Arciniegas et al. [23] mention the presence of cadinenolide, a hydrocarbon that produces essential oils, in *V. sphaerocephala*. The effects of these oils on germination depend on the concentrations at which they are applied and may not result in stimulating effects. Cerna-Chávez et al. [35] evaluated plant oils and extracts as forms of biocontrol with corn seeds and found that allelopathic effects were present, often observing a component–species–dose interaction. De Lillo et al. [36] reported that the application of essential oils from *Eucalyptus*, *Laurus*, and *Cinnamomum* spp. to *S. lycopersicum* seeds delayed germination without any subsequent negative effects on the seedlings.

Plant allelopathy is a ubiquitous ecological mechanism in nature and is an important factor that affects seed germination and sometimes harms plant growth; however, as already mentioned, many allelopathic effects will depend on the extract concentration. The higher the concentration of an extract, the more intense its inhibitory effect on seed germination, regardless of the species [16,37,38].

Some studies have reported allelopathic effects on seed germination using botanical extracts. Zhao et al. [39] reported an allelopathic effect of the aqueous extracts of sesame plants on bamboo seed germination and concluded that the observed effects were due to the amounts of phenols and alkaloids in sesame. Egushova et al. [40] reported that the allelopathic effects of the aqueous extracts of the weeds *Capsella* and *Sonchus* inhibited carrot germination. Rivero-Hernández [30] suggested that the application of aqueous extracts from *Melenis repens* to tomato and chili crops inhibits the development of radicles at high concentrations. Through liquid chromatography, Inderjit et al. [41] confirmed that the root leachate of *V. enceloides* contains phenolic compounds that have allelopathic effects in radish seeds.

With the results obtained in this study, we suggest that potential ranges of action of *V. sphaerocephala* and *V. fastigiata* may be expanded to inhibit the growth of weeds from different botanical families through the effects of certain molecules that they contain; however, more studies with chemical approaches are needed to narrow down the uses and applications of these biomolecules, as they may also function as growth promoters.

*4.2. Effect of V. sphaerocephala and V. fastigiata Extracts on Seedling Growth in S. lycopesicum and C. sativus*

The aqueous extracts from *V. sphaerocephala* and *V. fastigiata* significantly suppressed seed germination in *S. lycopersicum* and *C. sativus*, while seedling developmental responses to the aqueous extracts varied between developmental morphologies. The *V. sphaerocephala* extract affected the development of the root area in *S. lycopersicum* and *C. sativus*, whereas the *V. fastigiata* extract affected the root area in *S. lycopersicum* and the root length in *C. sativus*, which is supported by the results of [42,43].

Although statistically significant results were not found with regard to root area, root length was favorably affected in both *S. lycopersicum* and *C. sativus* with both *Verbesina* extracts, and both the leaf area and length were favored, which will continue to support sound physiological development (Figure 7a,b). This does not mean that a notable root area is optimal for the continued favorable development of the plant, but rather the number of

root hairs. Nutrients and aqueous extracts that cross the Caspary bands of root hairs into the xylem through a symplastic route are thus absorbed by the stem.

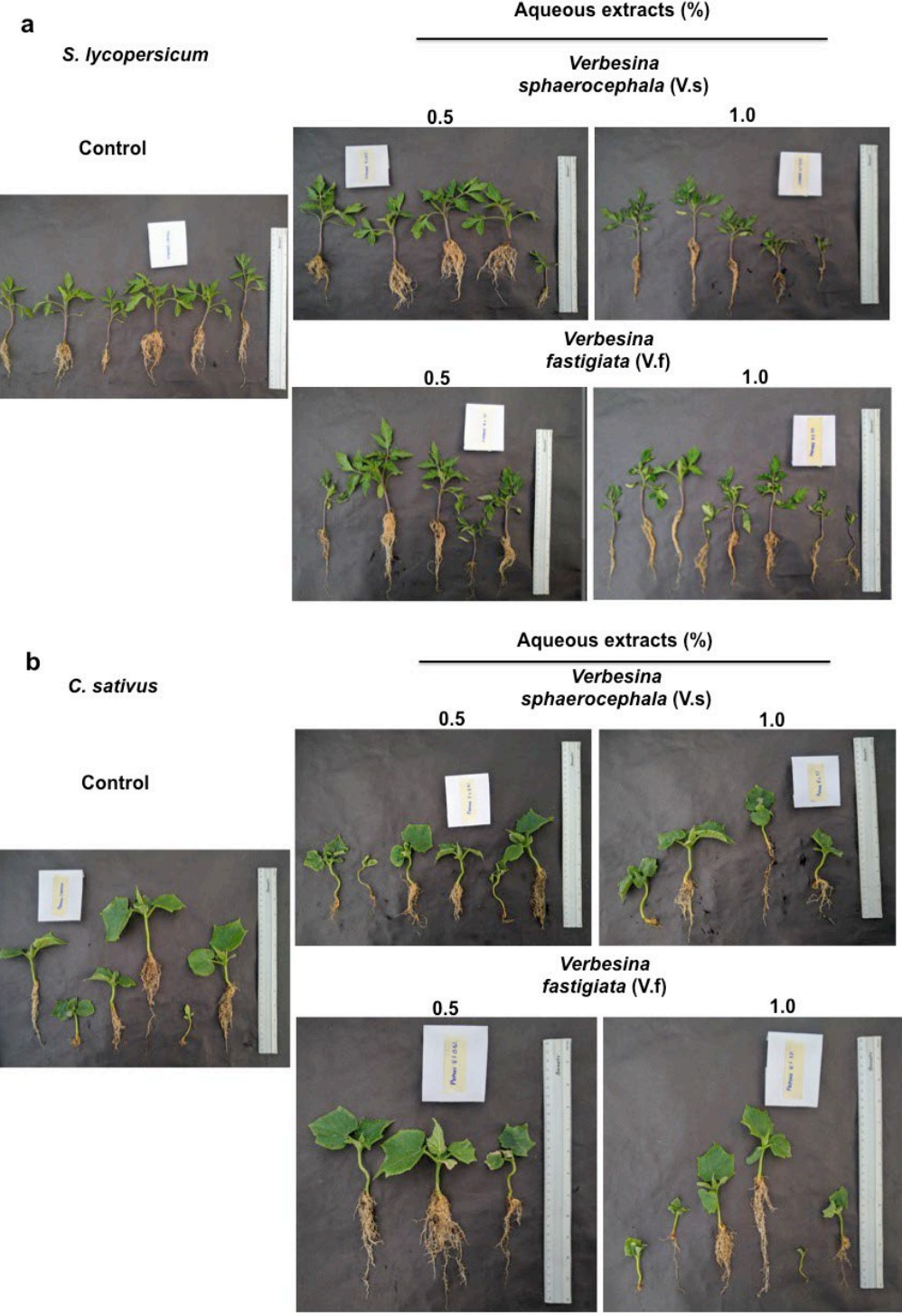

**Figure 7.** Differences in the development of seedlings in the nursery among treatments. Development of (**a**) *Solanum lycopersicum* and (**b**) *Cucumis sativus* seedlings with the application of extracts from *Verbesina sphaerocephala* and *Verbesina fastigiata* at concentrations of 0.5 and 1%.

The term biostimulant is relatively new, and its use in the scientific community is still being defined [44]. However, du Jardin [2] provides a broad definition: "A plant biostimulant is any substance or microorganism applied to plants with the aim of improving nutritional efficiency, tolerance to abiotic stress and/or quality characteristics of the crop,

regardless of its nutrient content." In this study, we were able to verify that biostimulation is both a holistic and integrative process that requires scientists to recognize and understand all of the variables that favor physiological process, while avoiding the problems and pitfalls that hinder it, including pests, diseases, weather, nutrition, and cultural practices, which agrees with what has been stated by Navarro [45].

Several studies have demonstrated optimal development in seedlings in terms of agronomic variables by evaluating the inhibition and/or delay of germination. For example, Wang et al. [15] revealed that the aqueous extracts of three herbs allelopathically inhibit the germination of lettuce, but promote seedling growth in low concentrations, which is similar to what was found in this study. Mira et al. [46] reported the allelopathic effect of *Pteridium aquilinum* on the germination and growth of economically important weeds. In addition, Li et al. [16] studied the effects of aqueous fig leaf (*Ficus carica* L.) extracts on seed germination and seedling growth in three medicinal species and found that these negatively affected germination, but promoted development.

The results of our study regarding the promotion of seedling growth due to the application of the aqueous extracts of *V. sphaerocephala* and *V. fastigiata* may be applicable to large-scale production systems; however, more studies are needed at the chemical level to identify the biomolecules responsible for this growth, which are necessary to design an integrative approach to cultivation.

## 5. Conclusions

The aqueous extracts of *V. sphaerocephala* and *V. fastigiata* delayed seed germination in *S. lycopersicum* and *C. sativus*, even at low concentrations. However, the application of the aqueous extracts from both *Verbesina* species in nursery conditions promoted seedling growth in the roots and leaves. The aqueous extracts of both *Verbesina* species also generated allelopathic effects with regard to germination, and thus additional field and chemical studies are needed to properly harness the biomolecules that these species contain, as they also act as plant growth biostimulators.

**Author Contributions:** Conceptualization, A.P.V.-R.; methodology, A.V.-R. and J.C.-E.; software, R.M.H.-H.; validation, S.F.V.-R., A.C.R.-A. and M.I.T.-M.; formal analysis, R.M.H.-H.; investigation, A.P.V.-R., A.V.-R. and J.C.-E.; resources, A.P.V.-R. and A.V.-R.; data curation, J.C.-E. and R.M.H.-H.; writing—original draft preparation, A.P.V.-R.; writing—review and editing A.P.V.-R., S.F.V.-R. and A.C.R.-A.; visualization, M.I.T.-M. and R.M.H.-H.; supervision, S.F.V.-R and A.C.R.-A. All authors have read and agreed to the published version of the manuscript.

**Funding:** This research received no external funding.

**Institutional Review Board Statement:** Not applicable.

**Data Availability Statement:** All data are available in the manuscript file.

**Acknowledgments:** The authors thank the division of agronomic sciences of CUCBA for the use of their facilities.

**Conflicts of Interest:** The authors declare no conflict of interest.

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
