# Peer review of "The Impact of Aqueous Extracts of Verbesina sphaerocephala and Verbesina fastigiata on Germination and Growth in Solanum lycopersicum and Cucumis sativus Seedlings"

_horticulturae, doi:10.3390/horticulturae8070652_

Round 1

Reviewer 1 Report

Interesting work on effects of plant extracts on different plant growth stages. It is a pity that there is no analysis done on organic content of the extracts. It might have provided some more information as to why the plant responded in the way they did. 

If you did not determine the concentration of phenolics in the extracts you should not be concluding the following: The allelopathic response that caused the extracts of V. sphaerocephala and V. fastigiata to  inhibit seed germination in the plants in this study points mainly to the quantity and quality of phenolic compounds. 

Please also check the information you cite, for example the production numbers for tomato from different Mexican states in 2020 do not add up to a total mentioned for the whole Mexico in the text but are much higher.

Were the plants only fertilised via leaf fertilisation?

Author Response

"PLEASE SEE THE ATTACHMENT"

Reviewer 2 Report

In this study Ana Paulina Velasco-Ramírez and colleagues examined the effects of aqueous extracts from Verbesina sphaerocephala and Verbesina fastigiata on the germination and growth of tomato and cucumber. Their results indicate that both extracts have a similar effect inhibiting the germination while acting as a biostimulant of the growth. Although known for other extracts from similar spp. this work provides a new description of the impact of these extract in the plants analyzed and, interestingly, it suggests a potential use of them as biostimulants. Taking this into account I think that the MS can be accepted after the authors address the following comments:

Lines 18-19: I do not think it is “The main problem”. Authors could use instead: “one of the main problems”.

Lines 23-24: Why the authors selected these concentrations? They should specify the rationale used for the concentration selection.

Line 228: Why authors did not used a wider range of concentrations? (at least a 10 times factor). Using a lower concentration could avoid negative impact of the extracts on germination while maintaining the biostimulant effect on growth?

Lines 294-295: What does the polynomial (or lineal) trend means? Does it have any biological meaning? Authors should expand on this or just take it out from the MS.

Lines 417-425: The Discussion first paragraph it is out of context. Authors should start this paragraph indicating what they found.

Figure 7: Authors should provide a higher resolution image for this figure.

Author Response

"PLEASE SEE THE ATTACHMENT"
